# Multivariable G-E interplay in the prediction of educational achievement

**Andrea G. Allegrini**[1]*, **Ville Karhunen**[2], **Jonathan R. I. Coleman**[1,3], **Saskia Selzam**[1], **Kaili Rimfeld**[1], **Sophie von Stumm**[4], **Jean-Baptiste Pingault**[1,5], **Robert Plomin**[1]

**1** Social, Genetic and Developmental Psychiatry Centre, Institute of Psychiatry, Psychology and Neuroscience, King's College London, United Kingdom, **2** Department of Epidemiology and Biostatistics, School of Public Health, Imperial College London, United Kingdom, **3** NIHR Maudsley Biomedical Research Centre, King's College London, United Kingdom, **4** Department of Education, University of York, United Kingdom, **5** Division of Psychology and Language Sciences, University College London, United Kingdom

* andrea.allegrini@kcl.ac.uk

**Data Availability Statement:** The dataset used to produce the results described in this manuscript will be made available, subject to a suitable data sharing agreement, in compliance with (EU)

## Abstract

Polygenic scores are increasingly powerful predictors of educational achievement. It is unclear, however, how sets of polygenic scores, which partly capture environmental effects, perform jointly with sets of environmental measures, which are themselves heritable, in prediction models of educational achievement. Here, for the first time, we systematically investigate gene-environment correlation (rGE) and interaction (GxE) in the joint analysis of multiple genome-wide polygenic scores (GPS) and multiple environmental measures as they predict tested educational achievement (EA). We predict EA in a representative sample of 7,026 16-year-olds, with 20 GPS for psychiatric, cognitive and anthropometric traits, and 13 environments (including life events, home environment, and SES) measured earlier in life. Environmental and GPS predictors were modelled, separately and jointly, in penalized regression models with out-of-sample comparisons of prediction accuracy, considering the implications that their interplay had on model performance. Jointly modelling multiple GPS and environmental factors significantly improved prediction of EA, with cognitive-related GPS adding unique independent information beyond SES, home environment and life events. We found evidence for rGE underlying variation in EA (rGE = .38; 95% CIs = .30, .45). We estimated that 40% (95% CIs = 31%, 50%) of the polygenic scores effects on EA were mediated by environmental effects, and in turn that 18% (95% CIs = 12%, 25%) of environmental effects were accounted for by the polygenic model, indicating genetic confounding. Lastly, we did not find evidence that GxE effects significantly contributed to multivariable prediction. Our multivariable polygenic and environmental prediction model suggests widespread rGE and unsystematic GxE contributions to EA in adolescence.

## Author summary

Our study investigates the complex interplay between genetic and environmental contributions underlying educational achievement (EA). Polygenic scores are becoming increasingly powerful predictors of EA. While emerging evidence indicates that polygenic

General Data Protection Regulations. Requests for the data used in this study should be made to the Twins Early Development Study (TEDS), and is not subject to the general TEDS data access policy: http://www.teds.ac.uk/researchers/teds-data-access-policy.

**Funding:** TEDS is supported by a programme grant to RP from the UK Medical Research Council (MR/M021475/1 and previously G0901245), with additional support from the US National Institutes of Health (AG046938). The research leading to these results has also received funding from the European Research Council under the European Union's Seventh Framework Programme (FP7/2007- 2013)/grant agreement n˚ 602768 and ERC grant agreement n˚ 295366. RP is supported by a Medical Research Council Professorship award (G19/2). AGA and VK have received funding from the European Union's Horizon 2020 research and innovation programme under the Marie Sklodowska-Curie grant agreement no. 721567. JRIC is supported in part by the UK National Institute for Health Research (NIHR) as part of the Maudsley Biomedical Research Centre (BRC). This study represents independent research partly funded by the NIHR BRC at South London and Maudsley NHS Foundation Trust and King's College London. The views expressed are those of the authors and not necessarily those of the NHS, the NIHR or the Department of Health and Social Care. K.R. is supported by a Sir Henry Wellcome Postdoctoral Fellowship. SvS is supported by a Jacobs Fellowship (2017-2019) and a Nuffield award (EDO/44110). High performance computing facilities were funded with capital equipment grants from the GSTT Charity (TR130505) and Maudsley Charity (980). The funders had no role in study design, data collection and analysis, decision to publish, or preparation of the manuscript.

**Competing interests:** The authors have declared that no competing interests exist.

scores are not pure measures of genetic predisposition, previous quantitative genetics findings indicate that measures of the environment are themselves heritable. In this regard it is unclear how such measures of individual predisposition jointly combine to predict EA. We investigate this question in a representative UK sample of 7,026 16-year-olds where we provide substantive results on gene-environment correlation and interaction underlying variation in EA. We show that polygenic score and environmental prediction models of EA overlap substantially. Polygenic scores effects on EA are partly accounted for by their correlation with environmental effects; similarly, environmental effects on EA are linked to polygenic scores effects. Nonetheless, jointly considering polygenic scores and measured environments significantly improves prediction of EA. We also find that, although correlation between polygenic scores and measured environments is substantial, interactions between them do not play a significant role in the prediction of EA. Our findings have relevance for genomic and environmental prediction models alike, as they show the way in which individuals' genetic predispositions and environmental effects are intertwined. This suggests that both genetic and environmental effects must be taken into account in prediction models of complex behavioral traits such as EA.

## Introduction

Education is compulsory in nearly all countries because it provides children with the skills, such as literacy and numeracy, that are essential for successfully participating in society. How well children perform at school, indicated by their educational achievement (EA; not to be confused with educational attainment, which is a measure of years spent in education), predicts many important life outcomes, especially further education and occupational status [1]. Quantitative genetic research based on twin studies showed that EA is 60% heritable throughout the school years [2, 3]. These studies also suggested that about 20% of the variance of EA and other learning-related traits can be ascribed to shared environmental factors, for example growing up in the same family and going to the same school. However, the picture became more complicated with the discovery that ostensible measures of the environment associated with educational achievement showed genetic influence–most notably, parents' educational attainment, socio-economic status (SES) and aspects of the home environment [4].

Quantitative genetic theory distinguishes two types of interplay between genetic and environmental effects, genotype-environment correlation (rGE) and genotype-environment interaction (GxE) [5]. rGE occurs when an individual's genotype covaries with environmental exposures. There are three types of rGE: passive, active and evocative. Passive rGE results from the inheritance of both genetic propensities and environments linked to parental genotypes. That is, individuals inherit from parents a genetic predisposition to a particular trait, but parental genotypes are also associated with rearing environments that, in turn, increase the likelihood of developing a particular trait. For example, individuals with stronger genetic predispositions to educational attainment tend to grow up in higher socioeconomic status families [6]. Evocative rGE happens when individuals' genetic propensities evoke a response from the surrounding environment; for example children's predisposition to higher food intake might elicit restrictive food behaviors from their parents [7]. Active rGE results from individuals actively selecting environments that are linked to their genetic propensity; for example, individuals with a higher genetic predisposition to educational attainment tend to migrate to economically prosperous regions that offer greater educational opportunities [8].

GxE, on the other hand, refers to genetic moderation of environmental effects. That is, when the effects of environmental exposures on phenotypes depend on individuals' genotypes. Equivalently, environmentally moderated genetic effects occur when genetic effects on a phenotype depend on environmental exposures. Importantly, however, rGE may confound GxE effects [9]. For example, if a genetic predisposition for a particular trait is found in a particular environment, it is difficult to know whether this represents rGE between the trait and the environment or true GxE. This picture becomes even more complicated when we consider that environments are themselves heritable [4].

Research on GxE was rejuvenated when it became possible to include measured genetic and environmental factors in statistical models. Hundreds of studies were published purporting to show interactions between candidate genes and environmental measures as they predict behavioural traits. For example, a seminal GxE study in the field [10] showed that carriers of two copies of the short serotonine allele on the 5HTT gene exposed to adversity had an increased the risk for depression compared to their genetic counterpart. However, GxE effects such as these have a poor replication history [11, 12]. The main problem with this approach is that it ignores the high polygenicity of complex traits, with a reductionist focus on single 'candidate' variants. This combined with typically small sample sizes, underpowered to detect the very small effects that can be expected for GxE, led to a replication failure [13].

In complex traits, very few individual variants capture more than a tiny fraction of trait variance [14]. Genome-wide polygenic scores (GPS) are the missing piece for investigating the interplay between genes and environment because they can theoretically capture genetic influences up to the limit of SNP-based heritability, which is usually 25–50% of the total heritability for behavioural traits. GPS are indices of an individual's genetic propensity for a trait and are typically derived as the sum of the total number of trait-associated alleles across the genome, weighted by their respective association effect size estimated through genome-wide association analysis [15]. A GPS derived from a genome-wide association study of educational attainment (years of schooling) [16] predicts up to 15% of the variance of EA [17]. As more powerful GPS become available, they have begun to be used widely in research on GxE [18–23] and rGE [7, 24–27].

Recently it has been possible to dissect the role of parental genetics on child achievement by splitting the parental genome into transmitted alleles (indexing passive rGE) and non-transmitted alleles (indexing environmentally transmitted parental genetic effects). The latter demonstrated that parental genotypes are associated with the environment they provide for the child [28, 29]. In fact, a growing body of evidence is showing the importance of considering gene-environment correlation when assessing polygenic effects on trait variation [30, 31], especially for educationally relevant traits. Paralleling previous findings from the quantitative genetics literature, a key point is that environmental measures are themselves heritable and GPS effects can be mediated by the environment, while environmental effects can be accounted for by genetics (genetic confounding). In this sense, polygenic scores for cognitive traits are not pure measures of genetic predisposition: their predictive power also captures environmental effects. For the same reason, environmental measures are not pure measures of the environment.

Rather than examining rGE and GxE for single polygenic scores and environmental measures, here we look at sets of GPS [32] and environmental measures. A multivariable approach is especially warranted for EA because twin analyses show that the high heritability (60%) of EA reflects many genetically influenced traits, including personality and behaviour problems in addition to cognitive traits [33, 34]. Correspondingly, EA GPS is associated with a wide range of traits, including psychiatric, anthropometric and behavioural traits [35]. Similarly, environmental predictors of EA are also intercorrelated (e.g. SES and home environment).

However, it is not yet clear how sets of polygenic scores, partly capturing environmental effects, perform jointly with sets of environmental measures, which are themselves heritable, and the effect that their interplay (rGE and GxE) might have on prediction.

Here for the first time we systematically investigate the interplay of GPS and environmental measures in the multivariable prediction of tested educational achievement. We jointly analyse multiple GPS and multiple environmental measures, considering the effect of their interplay in hold-out set prediction. Specifically, we test the joint prediction of 20 well-powered GPS for psychiatric, cognitive and anthropometric traits and 13 proximal and distal measured environments including life events, home environment and SES (see methods for descriptions of all measures). First, we model polygenic scores (henceforth G model) and environmental measures (henceforth E model), separately and jointly (full model), to predict educational achievement in penalized regression models [36] with hold-out set tests of prediction accuracy. Models are tuned using repeated cross-validation in 80% of the sample and tested in the remaining 20% hold-out set. Penalized methods are especially warranted when dealing with multiple correlated predictors as they can overcome problems of multicollinearity and overfitting. To investigate the relative contributions of the employed predictors to the full model, we carry out post-selection estimation [37] of partial regression coefficients, testing independent effects of single GPS and environmental measures. Secondly, we separate direct from mediated effects of the multivariable G and E models on EA and assess rGE defined in terms of the GPS and environmental measures employed. Finally, we assess GxE using a hierarchical group-lasso technique [38] to systematically discover two-way interactions between all GPS and environmental measures, and test their improvement in prediction of EA.

## Results

### Joint modelling of GPS and environmental effects

In a first step we tested three models for association with EA: all genetic factors (polygenic scores; G model), all environmental factors (measured environments; E model), and a joint model of all factors (full model; G+E). The G+E model achieved the best hold-out sample prediction compared to the G or E models considered separately. The full model predicted 36% of the variance (95% CI = 30.4, 41.6) in EA (Fig 1 panel B, S2 Table), 6% more than the E model (30.1%; 95% CI = 24.3, 35.6; S1 Fig) and up to 18% more compared to the G model alone (18.3%; 95% CI = 12.7, 23.6; S2 Fig). Nested comparisons of the G+E model vs the G and E models separately indicated that the difference in hold-out set prediction accuracy between models (Fig 1 panel D, S2 Table) was significant for both the G+E model vs the E model (median $R^2$ diff = 5.9%; 95% CI = 2.8, 9.1) and the G+E model vs the G model (median $R^2$ diff = 17.7%; 95% CI = 13.2, 22.3). This suggested the presence of genetic effects on EA not mediated via environmental effects, and vice versa of environmental effects not accounted for by the genetic effects. Next, we untangled the specific independent contributions of GPS and measured environments to variation in EA.

### Best-model and coefficient estimation

The best G+E model selected via 10-fold repeated (100 repeats; Fig 1 panel A) cross-validation in the training set included 24 predictors, 14 of which were GPS (blue) while 10 were environments (orange) (Fig 1, panel C). Of these top EA-increasing variables were SES in early life, followed by the GPS for educational attainment (EA3 GPS) and the GPS for intelligence (IQ3 GPS), while the top trait decreasing variable was chaos at home at age 12. In terms of coefficient estimation, partial regression coefficients in post-selection inference analyses (Fig 2 and S3 Table) showed that EA3 GPS ($\beta$ = 0.13; 95% CI = 0.09, 0.17; p = 8.45E-7) and IQ3 GPS ($\beta$ =

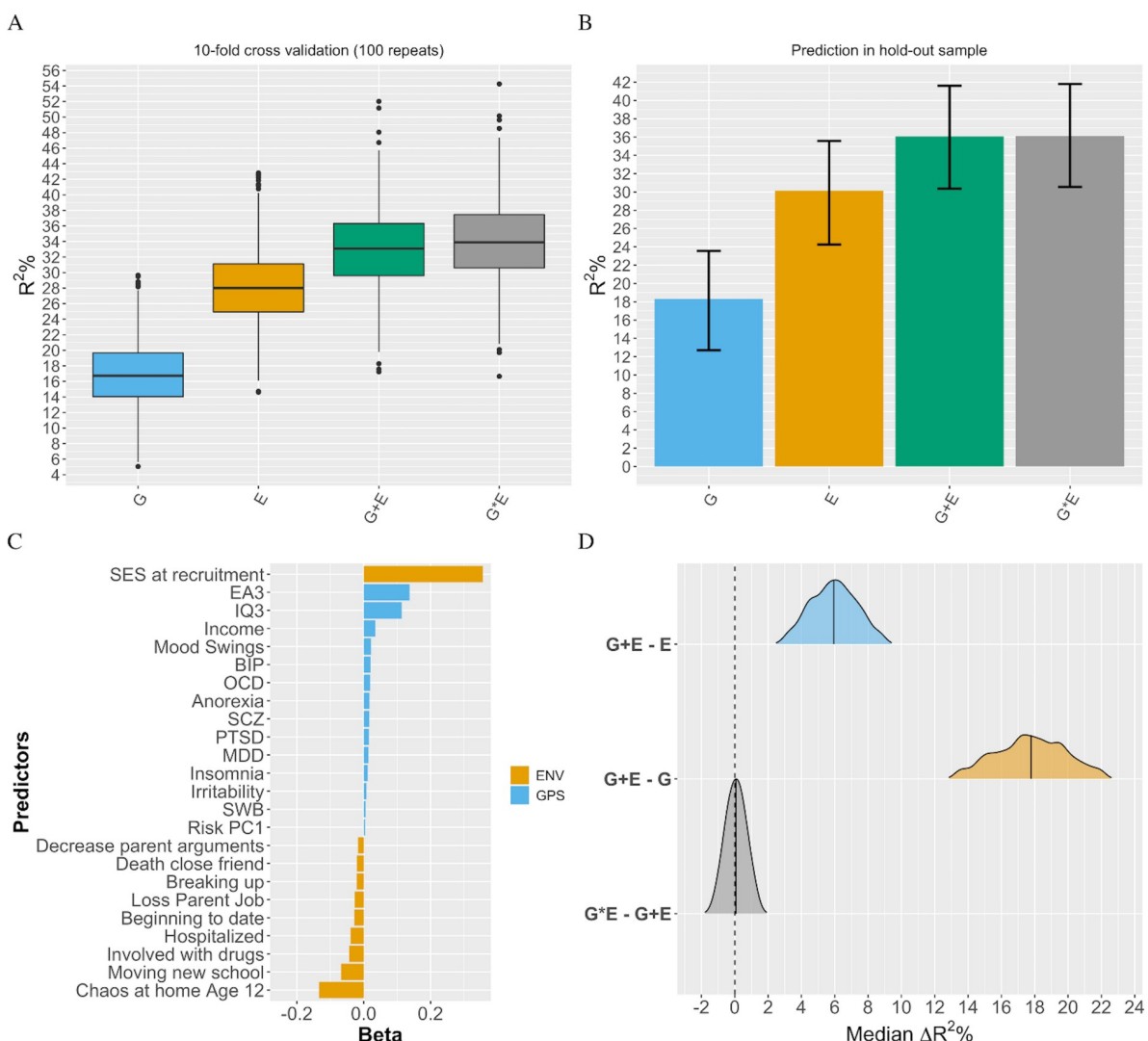

**Fig 1. Multivariable prediction of educational achievement. Panel A** = repeated 10-fold cross validation in training set, for the environmental (E), multi-polygenic score (G), joint (G+E), and interaction (G*E) prediction models. **Panel B** = Hold-out set prediction of EA for best models obtained via repeated cross validation in training set. Error bars are 95% bootstrapped confidence intervals. **Panel C** = G +E model used in hold-out set prediction. Figure shows variables selected via repeated cross-validation in the training set, and relative importance. **Panel D** = Comparison of prediction accuracy for models tested as bootstrapped $R^2$ difference between nested models in the hold-out set. Distributions represent independent (non-mediated) genetic effects (G+E−E), environmental effects (G+E−G), and G*E effects (G*E−G+E). **Note.** PGS = polygenic scores, ENV = Environmental measures. ASD = Autism Spectrum Disorder, BIP = Bipolar Disorder, BMI = Body Mass Index, EA3 = Educational Attainment, IQ3 = Intelligence, OCD = Obsessive Compulsive Disorder, PTSD = Post-Traumatic Stress Disorder, SCZ = Schizophrenia.

0.12; 95% CI = 0.08, 0.15; p = 1.33E-7) remained significant in the model after adjusting for the other predictors. SES was by far the most powerful predictor in the conditional model ($\beta$ = 0.37; 95% CI = 0.34, 0.40; p = 2.30E-60). Other environmental exposures that remained significant were 'chaos at home' at age 12 ($\beta$ = -0.14; 95% CI = -0.17, -0.12; p = 3.93E-15) and two life events experienced in the past year (all trait decreasing), including 'moving to a new school' ($\beta$ = -0.07; 95% CI -0.10, -0.04; p = 2E-5) and 'involved with drugs' ($\beta$ = -0.06; 95% CI -0.09, -0.03; p = 2E-3). SES, EA3 GPS, IQ3 GPS and 'chaos at home' were significant in all three models (i.e. naive, hold-out and conditional).

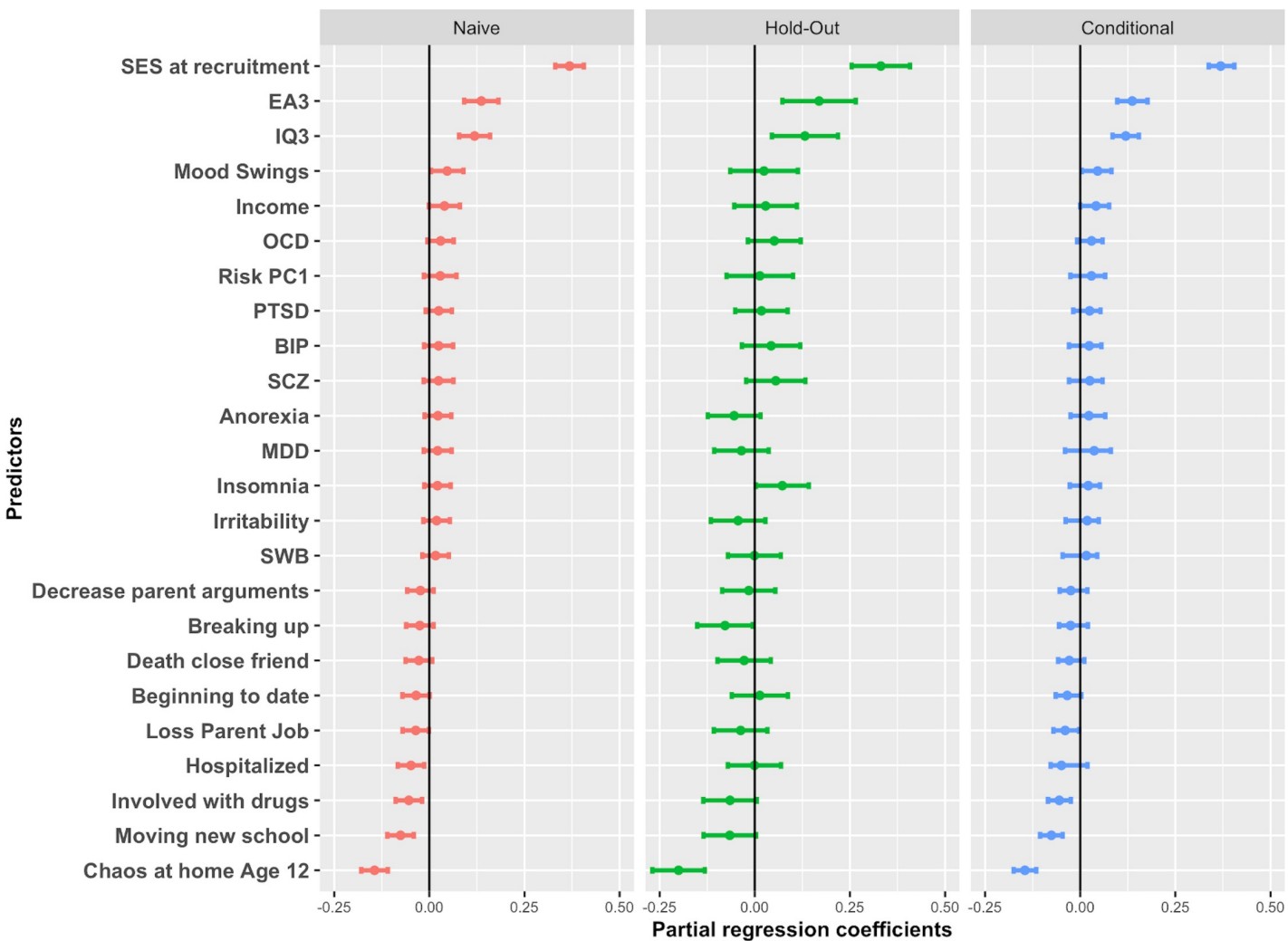

**Fig 2. Relative contributions of model selected variables for the G+E model in the prediction of educational achievement.** Figure shows partial regression coefficients, and 95% CIs around estimates. Naive = partial regression coefficients from multiple regression of selected variables in Training set; Hold-out = partial regression coefficients of selected variables in the hold-out set; Conditional = partial regression coefficients of training set for selected variables estimated with a conditional probability from a truncated distribution (see methods section). **Note.** ASD = Autism Spectrum Disorder, ADHD = Attention-Deficit Hyperactivity Disorder, BIP = Bipolar Disorder, EA3 = Educational Attainment, IQ3 = Intelligence, MDD = Major Depressive Disorder, SWB = Subjective Well-Being, OCD = Obsessive Compulsive Disorder, PTSD = Post-Traumatic Stress Disorder, Risk PC1 = First Principal Component of Risky behaviors, SCZ = Schizophrenia.

### rGE and mediated environmental vs GPS effects

S2 Table shows prediction model estimates for all models considered, as well as nested comparisons of hold-out set prediction accuracy ($R^2$) for the full model vs. E and the full model vs. G. We tested the correlation between the EA predicted values from the G model ($G_{ea}$) and the E model ($E_{ea}$) in the hold-out-set. This was r = 0.38 (95% CIs = 0.30, 0.45), indicating the extent of overlapping information between the G and E models in hold-out set prediction or, in other words, of rGE (as defined by the variables employed) underlying variation in EA. Then we proceeded to test the extent to which G and E effects on EA were reciprocally mediated (see methods). S4 Table shows results of mediation analyses. We found evidence for environmentally mediated genetic effects (indirect path: $\beta$ = 0.17; bootstrapped 95% CI 0.13, 0.21) and genetically mediated environmental effects (indirect path: $\beta$ = 0.10; bootstrapped

95% CI 0.06, 0.14). The effects of $G_{ea}$ on EA ($\beta$ = 0.43; bootstrapped 95% CIs = 0.36, 0.50) were reduced by 40% after introduction of the $E_{ea}$ mediator in the model ($\beta$ = 0.26; bootstrapped 95% CIs = 0.19, 0.33); these effects can be interpreted as the direct G model contributions to EA not accounted for by the E model. In other words, 40% of G effects on EA were explained by environmental mediation. Similarly, the direct $E_{ea}$ effects on EA ($\beta$ = 0.55; bootstrapped 95% CIs = 0.50, 0.60) were subject to a reduction of 18% ($\beta$ = 0.45; bootstrapped 95% CIs = 0.39, 0.51) after introduction of $G_{ea}$ as a mediator in the model, indicating partial genetic mediation of environmental effects (i.e. genetic confounding).

## GxE effects and multivariable prediction

We finally tested all possible two-way interactions jointly modelled by means of a hierarchical group lasso procedure using glinternet. Out of the possible 528 two-way interactions between all study variables (i.e. interactions between and within sets of GPS and environmental measures), 32 two-way interactions were detected by the hierarchical group-lasso technique (glinternet, S5 Table), 15 of which were GxE interactions. Fig 3 depicts an interaction network from the trained glinternet model (10-fold cross validation). Hold-out set prediction accuracy was only slightly improved ($R^2$ = 36.4%; 95% CI = 29.7, 41.1) over the joint G and E model ($R^2$ = 36.1%; 95% CI = 30.4, 41.6). We then introduced the 15 GxE interactions found in the full elastic net model (S3 Fig) to test whether they improved the prediction of EA over the full model that had only considered additive effects of GPS and environmental measures. There was no improvement in hold-out set prediction accuracy ($R^2$ = 36.1%; 95% CI = 30.5, 41.8), and the difference in prediction with the G+E model was not significant (median $R^2$ diff = 0.1%; 95% CI = -1.2, 1.3). S2 Table shows fit statistics for the glinternet and elastic net models. S5 Table reports GxE interactions detected by the hierarchical lasso model.

## Discussion

We tested the joint prediction accuracy of sets of multiple environmental measures and polygenic scores in prediction models of educational achievement and considered the effect of their interplay on model performance. Three main findings emerged from our analyses. First, the joint modelling of multiple GPS and related environmental exposures improved the prediction of EA, consistent with theory [39]. Second, paralleling previous quantitative genetic findings, we found consistent evidence of rGE effects underlying variation in EA (rGE = 0.38; 95% CIs = 0.30, 0.48), with a substantial proportion of polygenic score effects mediated by the environmental effects (40%), and evidence for genetic confounding (18%). Lastly, we did not find evidence that GxE effects jointly contributed to the prediction of EA.

Our multivariable GPS model alone predicted 18.3% of the variance in EA. Integration of multiple polygenic scores in the same model can be expected to increase as sample size in genome-wide association studies (GWAS) increases [40]. Here we constructed GPS in lasso-sum [41] based on previous observations that lassosum tends to perform better than more conventional approaches [17, 41] for educationally relevant traits. However, other methods for GPS construction can be expected to yield similar results when considering multivariable GPS penalized approaches, with performance of the relative approaches likely to converge as accuracy of GWAS estimates increases.

Previous work [16] showed that the predictive accuracy of EA3 provides unique information beyond correlated demographics and distal control variables such as income and parental educational attainment. Here we extend this observation to multiple polygenic scores within a prediction framework, as well as multiple measured environments, including proximal measures of home environment and life events. Furthermore, we take this a step further by

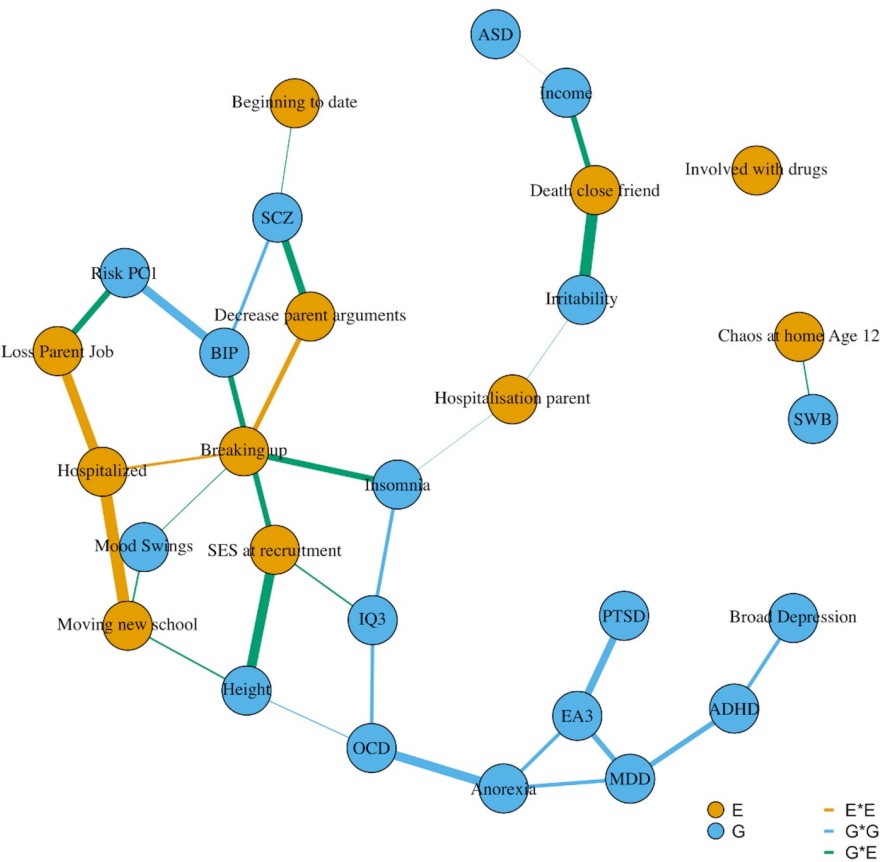

**Fig 3. Interaction network of glinternet model. Note.** Edges width represent interactions weights. E = Environmental measure, G = Genome-wide polygenic score. Polygenic scores acronyms: ASD = Autism Spectrum Disorder, ADHD = Attention-Deficit Hyperactivity Disorder, BIP = Bipolar Disorder, EA3 = Educational Attainment, IQ3 = Intelligence, Income = household income, MDD = Major Depressive Disorder, SWB = Subjective Well-Being, OCD = Obsessive Compulsive Disorder, PTSD = Post-Traumatic Stress Disorder, Risk PC1 = First Principal Component of Risky behaviors, SCZ = Schizophrenia.

formally testing trait associations of the relative polygenic scores and environmental measures when jointly considered in the same model.

Of interest were the relative contributions of the single GPS to the best model selected via repeated cross-validation in the training set. In post-selection inference analyses, IQ3 and EA3 were the only GPS independently associated with variation in EA after adjusting for measured environments and polygenic scores. This indicated that both these GPS contributed unique predictive information beyond other related, proxy environmental predictors (e.g. SES, parental educational attainment), and polygenic scores (e.g. household income GPS). Similarly, we found that several environments were independently predictive of EA. The best predictor was early life SES, a composite of parental educational attainment, employment status and maternal age at first birth. Life events and chaos at home were also significant contributors to the model, with negative independent effects on EA. Polygenic scores, however, improved the prediction of EA on top of the environment with a 20% increase in accuracy (from 30% to 36%). It is noteworthy that EA3 and IQ3 GPS were both significant in post-selection inference models after adjusting for SES, home environment and proximal environmental effects, all of which also tag genetic variance partly overlapping with that captured by the GPS. This suggested that cognitive-relevant GPS independently captured variation beyond environmental

variables and variance due to rGE in our model. While this was important to understand the model composition, it should be highlighted that these estimates are dependent on variables included in analyses, and can be expected to change as other variables are considered in the model (see below).

A central finding of the current study emerged when we separated direct and indirect effects of the GPS and environmental models by statistically testing for rGE. We found significant G mediation of the prediction of EA by the E model. This is in line with several quantitative genetics findings [42–44]. However, since it would be unreasonable to assume a causal effect of E on G (i.e. E does not change DNA sequence), in the sense employed here G acts as a 'confounder'–in causal modelling parlance, 'third variable confounding'–of E effects on EA ($E \leftarrow G \rightarrow EA$). That is, because our G model is associated with both the E model and EA, it partly induces an association between the E model and EA in addition to the independent effects of E on EA. This rGE effect explained 18% of the E effects on EA.

Different types of genetic confounding have been described in detail elsewhere [45].

Conversely, we also found evidence of environmental mediation of the G model effects on EA. The E model explained 40% of the GPS model effects on EA. This result is also in line with previous research in quantitative genetics [27–29, 46]. A growing body of evidence points to the rGE conclusion that genetic effects on cognitive trait variation are partly environmentally mediated [25], which is likely to be due to passive rGE. Passive rGE emerges because parents create a family environment that corresponds to their genotypes and, by extension, also correlates with the genotypes of their offspring. As previously described, alternative mechanisms include evocative and active rGE effects. As noted elsewhere [26] these possibilities are not mutually exclusive. However, in order to disentangle these rGE effects, different study designs are needed, for example, looking within families at the effects of maternal and paternal non-transmitted genotypes on child outcomes. Disentangling the different underlying mechanisms to the predicted variance in this regard is an issue for future studies, but out of the scope of the present investigation.

Previous work [30] has shown that prediction of educational achievement by EA3 GPS consistently decreases within-family, suggesting that passive gene-environment correlation explains part of the predictive power of EA3. Here we show that reciprocal indirect effects between multivariable E and G models explain a substantial proportion of variation of their total effects on EA. These results provide converging evidence with recent research looking at rGE underlying parenting and children educational attainment [27, 47]. Both genetic confounding and environmental mediation are important factors to take into account in the prediction of EA.

Lastly, we applied a hierarchical group-lasso model (glinternet) to automatically detect two-way interactions. This model helped us to identify GxE effects that show strong hierarchy, which would have otherwise been difficult to detect due to the great multiple-testing burden relative to the sample size of the present study. Furthermore, since glinternet performs shrinkage and grouping before testing for interaction effects, this enabled discovery of interactions that would have been confounded by strong main effects of correlated predictors. In other words, because the coefficients of main effects have been regularized (that is shrunk, see Methods), their fit is reduced, which facilitates the discovery of interaction effects [38]. However, neither the glinternet model including all discovered pairwise interactions, nor the elastic net model including two-way GxE effects, significantly improved hold-out set prediction over the G+E model. One possible explanation for this finding is that GxE effects are typically very small, and that the trade-off between true effect and variance introduced in the model, signal to noise ratio, was too small. It might be that even if two-way GxE effects were relevant the noise incurred in fitting their coefficients may outweigh the improvement in accuracy that

they bring to the model. In this regard we note that in repeated cross-validation in the training set the model performance of both the elastic net based GxE model, and the glinternet model was substantially increased compared to the G+E model. Application of this method in larger datasets, or using different phenotypes with different genetic architectures, might be fruitful for hypothesis-free GxE discovery as well as for prediction.

This study must be considered in light of a few limitations. First, our results are subject to the constraint that we performed an apriori selection on variables to be employed in our analyses. For example, we modelled exposures that are typically defined as environmental; however, many other variables can be argued to capture EA relevant environmental influences. In this regard estimates for non-mediated genetic effects for the model presently tested are likely upper-bounds, in the sense that if we were to include more E variables predictive of the outcome EA, the polygenic score contributions independent of E would either stay the same or decrease. Likewise, we included a broad range of polygenic scores that are currently available as the most predictive for cognitive, psychiatric and anthropometric traits. However, polygenic scores predictive power is in part a function of GWAS sample size [40], therefore as more powerful GWAS become available these prediction estimates are expected to increase. This in turn suggests that the contributions of the G model are likely to be on the lower bound compared to future polygenic score work in this area.

We note that the environmental variables employed in the models jointly represent only a noisy proxy for the true EA relevant environmental effects, just as the multi-polygenic scores model is a noisy proxy for the true additive genetic predisposition to EA. As such, estimates derived from mediation analyses will imprecisely capture the extent to which E and G models are reciprocally mediated.

Finally, we focused here on EA but predictive models of other complex traits are likely to yield different results, because EA shows comparatively great shared environmental influences [30]. This suggests that rGE is likely to be stronger for EA than for other behavioural traits, such as personality traits and social-emotional competencies. Regarding our analytical approach, we focused on GxE interactions that obeyed strong hierarchy as identified by the group lasso technique. Future studies could relax this assumption and include interactions where one of the main effect sizes is not significant, as well as higher order interactions. Finally, although it is a strength of our study that we used measured environmental exposures, we note that methods for inferring GxE without measured environmental data are emerging that have reported GxE for some complex traits [48]. The extent to which these effects are systematic, stable, and generalizable to EA remains to be determined.

As large multidimensional biobank datasets become increasingly available, the integration of multi-omics data with multiple environmental measures will become more common in prediction modelling. Here, we provide an indication of the effects of integrating multiple GPS and environmental measures in prediction models of EA and the effect that their interplay has on prediction accuracy in a population cohort of adolescents. In conclusion, we found consistent evidence for rGE in prediction models of EA that systematically tested the interplay between polygenic scores and measured environments within a hypothesis-free multivariable prediction framework. When integrating multiple GPS and environmental measures, their interplay must be taken into account. Separate effects of environmental and polygenic scores cannot just be assumed to add up because pervasive rGE affects prediction.

## Material and methods

### Sample

We test our models using data from 16 year olds from the UK Twin Early Development Study [TEDS; 49], a large longitudinal study involving 16,810 pairs of twins born in England and Wales between 1994–1996, with DNA data available for 10,346 individuals (including 3,320 dizygotic twin pairs and 7,026 unrelated individuals, all of European ancestry). Genotypes for the 10,346 individuals were processed with stringent quality control procedures followed by SNP imputation using the Haplotype Reference Consortium (release 1.1) reference panels. Current analyses were limited to the genotyped sample, and we retained only one individual at random from sibling pairs. Following imputation, we excluded variants with minor allele frequency <0.5%, Hardy-Weinberg equilibrium p-values of $<1 \times 10-5$. To ease computational demands, we selected variants with an info score of 1, resulting in 515,000 SNPs used for analysis (see S1 Methods for a full description of quality control and imputation procedures).

### Ethics statement

Ethical approval for TEDS has been provided by the King's College London Ethics Committee (reference: PNM/09/10–104). Written parental consent was obtained before data collection.

### Measures

**Dependent measure.**   Educational achievement was measured as the self-reported mean grade of three core subjects (English, math and science) scored by the individuals at age 16 in their standardized UK General Certificate of Secondary Education (GCSE) exams.

EA was operationalized as the mean grade of the three compulsory subjects, with results coded from 4 (G, or lowest grade) to 11 (A+, or highest grade). These self-report measures are highly replicable and show high genetic and phenotypic correlations with teacher scores [50]. The variable distribution was slightly negatively skewed (similar to the national average) and subject to a rank based inverse normal transformation to approximate a normal distribution.

**Environmental measures.**   Socio economic status: SES at recruitment (mean children age = 18 months) was calculated as the mean composite score of five standardized measures including mother and father qualification levels ranging from 1 = 'no qualifications' to 8 = 'postgraduate qualification', mother and father employment status [51], and mother's age at birth of first child.

Chaos at home: as a measure of home environment a shortened version of the Confusion, Hubbub and Order Scale [52] was used to measure children's perception of chaos in the family environment at age 12. Children rated the extent to which they agree (range: 'not true', 'quite true' or 'very true') to six items: 'I have a regular bedtime routine' (reversed coded), 'You can't hear yourself think in our home', 'It's a real zoo in our home', 'We are usually able to stay on top of things' (reversed coded), 'There is usually a television turned on somewhere in our home' and 'The atmosphere in our house is calm' (reverse coded). The Chaos score was computed as the mean of the rated items.

Life events: Self-reported life events experienced in the past year were measured (at age 16) using a shortened version of the Coddington life events [53]. Individuals had to report on 20 items that might have happened in the past year, by responding yes (coded as 1) if the event had happened or no (coded as 0) if it didn't happen. Items included stochastic, proximal events such as "death of a close friend or relative", "being hospitalized", as well as family-wide events e.g. "loss of a parent job", "decrease in parental income". When considering prediction of educational achievement, educationally relevant items were removed from the models (i.e. "failing

exam" and "outstanding achievement"). Items being endorsed by fewer than 100 people were discarded from analyses. A total of 11 life events were retained in analyses. All items were considered separately in prediction models (i.e. they were not aggregated in a scale). S1 Table reports descriptive statistics for variables employed in this study, separately by training and hold-out sets.

**Genome-wide polygenic scores (GPS).** GPS for 20 cognitive, anthropometric and psychopathological traits were constructed using Lassosum [41]. Lassosum is a penalized regression approach applied to GWAS summary statistics. In Lassosum we try to minimize the following loss function:

$$y^T y + (1-s)\beta^T X_{ref}^T X_{ref}\beta - 2\beta^T r + s\beta^T \beta + 2\lambda \|\beta\|^1 \qquad (1)$$

Where y is a vector of the phenotype, X is the matrix of genotypes, such that $X_{ref}^T X_{ref}$ is a matrix of correlations between SNPs, the LD matrix. r denotes the correlation between SNPs and the phenotype, $r = X^T y$. The subscript $_{ref}$ in $X_{ref}^T X_{ref}$ indicates that SNPs employed to obtain the LD matrix (based on a reference panel, see below) will generally not correspond to SNPs used to infer the correlation with the phenotype.

In the equation $\lambda$ controls the L1 penalty (L1 norm, [54]). The notation $\|\beta\|^1$ describes the L1 norm of a coefficient vector $\beta$, defined as $\|\beta\|^1 = \Sigma|\beta|$, while $s$ is another tuning parameter controlling the L2 penalty ($|\beta|^2$, the sum of the squared betas). Here $s$ has the additional constraint of being between 0 and 1. When $\lambda = 0$ and $s = 1$ the problem becomes unconstrained.

Tuning parameters, $\lambda$ and s, are chosen in the validation step (this is akin to optimization that can be performed in p-value thresholding methods). We used our training set to perform parameter tuning optimizing (with respect to $R^2$) polygenic scores against EA. LD was accounted for via a reference panel, here the same as the training set sample, and estimation of LD blocks was performed using LD regions defined in [55].

We employed the most powerful and publicly available GWAS summary statistics for cognitive, psychopathology and anthropometric traits. We created cognitive and educationally relevant polygenic scores for educational attainment [16], intelligence [56], and income [57]. We also created polygenic scores for mental health-related traits: autism spectrum disorder [58], major depressive disorder [MDD; 59], bipolar disorder [BIP; 60], schizophrenia [SCZ; 61], attention deficit hyperactivity disorder [ADHD; 62], obsessive compulsive disorder [OCD; 63], anorexia nervosa [AN; 64], post-traumatic stress disorder [PTSD; 65], broad depression [66], mood swings [67], subjective well-being [68], neuroticism [69], irritability [67], insomnia [70], and risk taking [71]. Finally, we created polygenic scores for height and BMI [72]. S6 Table reports information on these summary statistics, while S7 Table reports parameter tunings for the lassosum GPS.

## Analyses

All variables were first regressed on age, sex, 10 genetic principal components and genotyping chip. The obtained standardized residuals were used in all subsequent analyses.

**Penalised regression.** We fit elastic net regularization [73] models for EA. Elastic Net minimizes the residual sum of squares (RSS) subject to the L1 penalty, consisting of the sum of the absolute coefficients, which introduce sparsity allowing for parameters selection, and the L2 penalty, consisting of the sum of the squared coefficients, which allows for parameters shrinkage [73].

Elastic net tries to minimise the following loss function:

$$\|y - X'\beta\|^2 + \lambda(\alpha^*|\beta|^1 + (1-\alpha)^*|\beta|^2) \qquad (2)$$

where $||y–X'\beta||^2$ is the residual sum of squares, $|\beta|^2$ is the sum of the squared betas (the L2 penalty), and $|\beta|^1$ is the sum of the absolute betas (the L1 penalty).

Here X' is an nxp ('n' observations and 'p' predictors) matrix of polygenic scores, environmental predictors or a combination of both (see below). $\alpha$ determines the mixing of penalties, where the first parameter introduces sparsity while the second shrinks correlated predictors towards each other. $\lambda$ is a tuning parameter that control the effect of the penalty terms over the regression coefficients. When $\alpha = 1$ the solution is equivalent to a LASSO regression, while when $\alpha = 0$ the solutions is equivalent to a Ridge regression. For every $\alpha$ multiple $\lambda$ exists, and the optimal combination of tuning parameters is determined by cross-validation, here a 10-fold cross-validation repeated 100 times. For every model tested we split the sample into an independent training set (80%) and a hold-out set (20%). In the training set we perform 10-fold repeated cross-validation to select the model that minimises the Root Mean Square Error (RMSE)–that is the tuning parameter for which the cross-validation error is the smallest. The model performance is then assessed by the variance explained ($R^2$) in the hold-out test set. The hold-out set $R^2$ was calculated as $1-\frac{SSE}{SST}$ (SSE = sum of squared errors, SST = sum of square total).

**Bootstrapping.**   For every model tested we sampled with replacement from the data (1000 times) to calculate bootstrapped confidence intervals for the hold-out set prediction accuracy ($R^2$). Rows of data for resampling included the phenotype under study and the predictors according to the model tested: either polygenic scores, environmental predictors or a combination of both. For each bootstrap sample drawn we calculated the hold-out set $R^2$, and we took the difference in $R^2$ between nested models. This procedure yielded a distribution of $R^2$ for each model tested and a distribution of $R^2$ differences ($\Delta R^2$) for each pairwise comparison. We then calculated 95% confidence levels as the 2.5th and 97.5th percentiles of these distributions. For nested comparisons, if the interval didn't contain 0 we concluded that the pairwise model $\Delta R^2$ was significantly different from 0 with a $\alpha$ level of .05.

**Post selection inference.**   For every model tested we conducted statistical inference of models coefficients after selection of most informative predictors performed by Elastic Net, that is effect sizes, p-values and confidence levels around the prediction estimates.

Post-selection inference [37] refers to the practice of attempting to accurately estimate prediction coefficients after a model selection has been performed. If we fit the optimal model's selected predictors in a multiple regression model in the training set (that is where the selection has been performed) our confidence in the estimates will tend to be over-optimistic. On the other hand, estimation of these parameters in a hold-out set would not be subject to this problem. The hold-out set, however, will typically be smaller than the training set, leading to wider confidence intervals. In addition, the results will be dependent on the random split (80–20) performed. A third way is to calculate P-values conditional to the selection that has been made in the training set. Briefly, after selection is performed, accurate estimation of a given partial regression coefficient can be approximated by a truncated normal distribution:

$$\widehat{\beta} \sim \mathrm{TN}^{a,b}(\beta, \tau^2) \tag{3}$$

With mean β, variance $\tau^2$ and boundaries of the truncated normal distribution (TN) 'a' and 'b' given by the data and the selection procedure, in this case the predictors, the active set (the variables with non 0 coefficients selected by our model) and $\lambda$ [37]. We refer elsewhere to a thorough discussion of the topic [74], with a focus on lasso like approaches. Here we compare results from the three procedures: the 'naive' estimation of partial regression coefficients in the training set, estimation of coefficients in the hold-out set, and the conditional estimation of p-values performed using the R package 'SelecitveInference' [75].

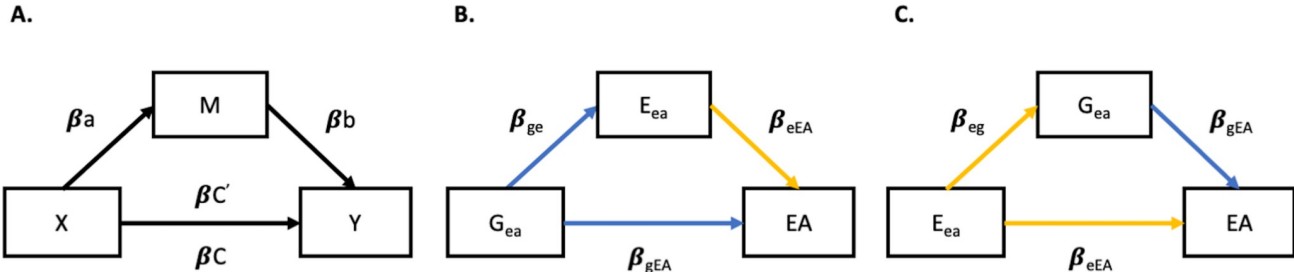

**Fig 4.** *Panel A* = schematic representation of mediation analysis; *βC* = effect of a predictor X on an outcome Y; *βa* = effect of X on a mediator (M); *βb* = effect of M on Y after adjusting for X; *βC'* = effect of X on Y after adjusting for M. *Panel B* = Directed acyclic graph (DAG) showing $E_{ea}$ mediated effects of $G_{ea}$ on EA in the hold out-set; $\beta_{ge}$ = causal path between $G_{ea}$ and $E_{ea}$ equivalent to $r_{G,E}$; $\beta_{eEA}$ = direct independent $E_{ea}$ effects on EA; $\beta_{gEA}$ = total $G_{ea}$ effects on EA. *Panel C* = DAG showing $G_{ea}$ mediated effects on EA (genetic confounding, see methods and discussion); $\beta_{eg}$ = causal path between $E_{ea}$ and $G_{ea}$ equivalent to $r_{G,E}$; $\beta_{gEA}$ = direct independent $G_{ea}$ effects on EA; $\beta_{eEA}$ = total $E_{ea}$ effects on EA. Note. Blue paths represent G model effects, yellow paths represent E model effects.

**rGE.** We quantified rGE in two ways. First, the hold-out set predicted EA values from the GPS (henceforth $G_{ea}$) and environmental (henceforth $E_{ea}$) models can be tested for correlation. In this sense the covariance between these variables would be an indication of overlapping information between E and G underlying EA, i.e. $r_{G,E}$ = cor($G_{ea}$,$E_{ea}$).

Second, another way to quantify rGE is by modelling E and G effects in a mediation model (Fig 4), considering the indirect effects of G on EA via E, and vice versa the indirect effects of E on EA via G. We used the predicted EA values from the GPS and environmental models (i.e. $G_{ea}$ and $E_{ea}$) to test mediation models in the hold-out set. We fit a structural equation model (SEM) in 'lavaan' [76] to test whether and to what extent E and G effects on EA were reciprocally mediated. Panel A (Fig 4) is a schematic representation of a mediation model, where βC is the effect of a predictor X on an outcome Y, βa the effect of X on the mediator (M), and βb the effect of M on Y after adjusting for X. βC' corresponds to the effects of the predictor on the outcome when controlling for the mediator (i.e. when the full equation is estimated). If the effects are reduced (partial mediation) or are not different from 0 (full mediation) then there is evidence for mediation. We quantify the proportion of the mediated effects as (βC- βC') / βC and test for significance of the indirect path using bootstrapping (with 1000 repetitions).

Fig 4 represent direct and indirect effects of the G model effects on EA mediated by E (panel B), and of the E model effects on EA mediated by G (panel C). While panel B represents a causal model where we estimate the environmentally mediated G effects on EA, panel C is a statistical abstraction since it would be unreasonable to assume a causal relationship of E on G. Here we model G as mediator to estimate the third variable confounding effects underlying the relationship between E and EA, as mediating and confounding effects have been shown to be equivalent in a linear context [77]. In other words, confounding and mediation effects are statistically equivalent, such that they can both be estimated by mediation analysis, but conceptually distinct (77).

**GxE.** After fitting the joint GPS and environmental models, we apply a hierarchical lasso procedure to automatically search the feature space for interactions, and retrain our models introducing GxE interactions. With 33 predictors there is a total of 33(33–1)/2 = 528 possible 2-ways interactions. Testing all models separately would imply a multiple testing burden (e.g. bonferroni correction .05/528 = 9E-5), in addition to the expected low signal to noise ratio for GxE effects. Here we employ a hierarchical group lasso approach to automatically search for two-way interactions, implemented in the R package 'glinternet' [38] (group-lasso interaction network). Glinternet leverages group lasso, an extension of LASSO, to perform variable selection on groups of variables, dropping or retaining them in the model at the same time, to select

interactions. As noted above, the L1 regularization produces sparsity. Glinternet uses a group lasso for the variables and variable interactions, which introduces a strong hierarchy: an interaction between two variables can only be picked by the model if both variables are also selected as main effects. That is, interactions between two predictors are not considered unless both predictors have non-zero coefficients in the model. Once two-way interactions obeying strong hierarchy were identified, we selected GxE interactions (i.e. GPS that interact with environmental variables) and reintroduced them in our best elastic net models to test whether the hold-out set prediction accuracy improved beyond the full (E+G) prediction model.

## Supporting information

**S1 Table. Descriptive statistics.**
(XLSX)

**S2 Table. Training vs hold-out set fit indices, and nested comparisons.**
(XLSX)

**S3 Table. Statistical inference.**
(XLSX)

**S4 Table. Mediation models, bootstrapped estimates and 95% Confidence Intervals.**
(XLSX)

**S5 Table. List of interactions identified through Glinternet.**
(XLSX)

**S6 Table. GWAS Summary statistics.**
(XLSX)

**S7 Table. Parameter tuning for lassosum GPS.**
(XLSX)

**S1 Methods. Quality control and genotyping protocol.**
(DOCX)

**S1 Fig. Polygenic score (G) model used in hold-out set prediction.** Variables importance for the best G model selected via repeated cross-validation in the training set. **Note.**
ASD = Autism Spectrum Disorder, ADHD = Attention-Deficit Hyperactivity Disorder, BIP = Bipolar Disorder, EA3 = educational attainment, IQ3 = intelligence, MDD = Major Depressive Disorder, SWB = Subjective Well-Being, OCD = Obsessive Compulsive Disorder, PTSD = Post-Traumatic Stress Disorder, SCZ = Schizophrenia.
(TIF)

**S2 Fig. Environmental predictors (E) model used in hold-out set prediction.** Variables importance for the best E model selected via repeated cross-validation in the training set.
(TIF)

**S3 Fig. G\*E model used in hold-out set prediction.** Variables importance for the best G\*E model selected via repeated cross-validation in the training set. **Note**. For interactions the first name refers to polygenic scores, the second name refers to environmental predictors.
ASD = Autism Spectrum Disorder, ADHD = Attention-Deficit Hyperactivity Disorder, BIP = Bipolar Disorder, EA3 = educational attainment, IQ3 = intelligence, MDD = Major Depressive Disorder, SWB = Subjective Well-Being, OCD = Obsessive Compulsive Disorder, PTSD = Post-Traumatic Stress Disorder, SCZ = Schizophrenia.
(TIF)

## Acknowledgments

We gratefully acknowledge the ongoing contribution of the participants in the Twins Early Development Study (TEDS) and their families.

## Author Contributions

**Conceptualization:** Andrea G. Allegrini, Robert Plomin.

**Data curation:** Andrea G. Allegrini, Saskia Selzam.

**Formal analysis:** Andrea G. Allegrini.

**Funding acquisition:** Robert Plomin.

**Investigation:** Robert Plomin.

**Methodology:** Andrea G. Allegrini, Ville Karhunen, Jonathan R. I. Coleman, Jean-Baptiste Pingault.

**Software:** Andrea G. Allegrini, Jean-Baptiste Pingault.

**Supervision:** Jean-Baptiste Pingault, Robert Plomin.

**Visualization:** Andrea G. Allegrini.

**Writing – original draft:** Andrea G. Allegrini, Robert Plomin.

**Writing – review & editing:** Andrea G. Allegrini, Ville Karhunen, Jonathan R. I. Coleman, Saskia Selzam, Kaili Rimfeld, Sophie von Stumm, Jean-Baptiste Pingault, Robert Plomin.

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
