## [Decision Letter · Decision Letter 0]

25 Jun 2020

Dear Dr Allegrini,

Thank you very much for submitting your Research Article entitled 'Multivariable G-E interplay in the prediction of educational achievement' to PLOS Genetics. Your manuscript was fully evaluated at the editorial level and by independent peer reviewers. The reviewers appreciated the attention to an important problem, but raised some substantial concerns about the current manuscript. Based on the reviews, we will not be able to accept this version of the manuscript, but we would be willing to review again a much-revised version. We cannot, of course, promise publication at that time.

In addition, PLOS Genetics policy requires that data be made available to other researchers, and that this availability not be at the sole discretion of authors. Therefore, we would like to have you clarify data sharing prior to a final decision on your manuscript.. 

If you decide to revise the manuscript for further consideration at PLOS Genetics, please aim to resubmit within the next 60 days, unless it will take extra time to address the concerns of the reviewers, in which case we would appreciate an expected resubmission date by email to plosgenetics@plos.org.

[LINK]

We are sorry that we cannot be more positive about your manuscript at this stage. Please do not hesitate to contact us if you have any concerns or questions.

Yours sincerely,

Chris Cotsapas, PhD

Associate Editor

PLOS Genetics

Scott Williams

Section Editor: Natural Variation

PLOS Genetics

Reviewer's Responses to Questions

**Comments to the Authors:**

Reviewer #1: Uploaded as an attachment

Reviewer #2: See attachment.

Reviewer #3: In this paper, the authors consider the joint role of genome-wide polygenic scores (GPSs) and environmental variables at predicting educational achievement. While it's known that such measures are known to be individually good predictors of educational variables, this study quantifies the degree of overlapping signal between a set of genetic and environmental variables. At a high level, they do this using regularized prediction models and mediation-style analyses. They find that 40% of the predictive power of GPSs and 18% of the predictive power of environmental variables is due to the correlation between these variables. They also test whether interactions of environmental and genetic variables add to the predictive power of joint genetic-environmental predictive models, and find no evidence that interactions of the variables they considered contribute significantly to multivariable prediction.

This paper represents an impressive effort in a rich data set. There is a lot of interesting information presented. Also the paper was easy to read, the analyses were clearly described in a way that I believe I would be able to replicate the analyses, and the tables and figures had clear and comprehensive captions. That said, upon reading the paper, I found myself with a lot of questions about what the major take-aways are for this project.

Major concerns:

1) How does this paper fit relative to other papers that do similar work? The authors make several strong statements about the novelty of their work, but there are a number of papers that ask very closely related questions. For example, Selzam et al (2019) compare the within- and between-family predictive power of GPSs for educational achievement, which would represent the passive rGE that the authors describe. Also, Lee et al. (2018) contains an analysis quite similar to this one looking at the incremental R2 of an education GPS at predicting educational attainment above several environmental variables. While educational attainment and educational achievement are different variables, they are very highly genetically correlated. It would be helpful if the authors could explicitly highlight what we are learning from this paper relative to related existing work.

2) I had a difficult time interpreting their rGE results for a number of reasons.

a) How does the sparse variable selection procedure affect the authors' estimates? As far as I could tell, the authors selected their genetic and environmental variables using an elastic net procedure in the same step. If there is strong rGE (as the authors find) some of the genetic variables that are highly correlated with the environmental variables may be selected in the place of the environmental variables in the penalization process, even if they represent independent signals. Likewise, some environmental variables will be selected in the place of the genetic variables. Since you are removing variables that are correlated with each other, this would have an effect on the subsequent rGE analysis. It may be possible to address this concern by selecting the genetic and environmental variables separately.

b) How appropriate is it to estimate the mediation analysis in both directions? In order to interpret the estimates of how much the environmental variables mediate the genetic variables, you need to assume the model in Figure 4B. To estimate the amount of genetic confounding, you need to assume the model described in line 324 of the manuscript. Since both of these models can't be simultaneously true, I believe it can't be the case that the two estimates in lines 35 to 39 are both true.

c) How do errors in variables affect the interpretation of these analyses? For example, the 'chaos at home' variable is likely a noisy proxy for the true amount of chaos at home. This means that even if 100% of the genetic signal were mediated through this variable, the GPS would still remain predictive after including that variable in the regression. Similarly, the polygenic scores may be thought of as noisy measures of the true additive genetic component. If that's the case, then a mediation analysis will overestimate the amount that the environmental variables mediate the predictive power of the genetic factor.

Minor concern:

3) The notation used the methods section is confusing. The authors often use the same variables to represent different things. For example, in the equation on line 463, the authors use 'r' to denote both a vector covariances and also as an indicator that the genotypes come from a reference sample. Also the variables and coefficients in line 463 correspond to different variables and parameters in line 505 despite having the same names.

References:

Lee, J. J., Wedow, R., Okbay, A., Kong, E., Maghzian, O., Zacher, M., ... & Fontana, M. A. (2018). Gene discovery and polygenic prediction from a 1.1-million-person GWAS of educational attainment. Nature Genetics, 50(8), 1112.

Selzam, S., Ritchie, S. J., Pingault, J. B., Reynolds, C. A., O’Reilly, P. F., & Plomin, R. (2019). Comparing within-and between-family polygenic score prediction. The American Journal of Human Genetics, 105(2), 351-363.

**Have all data underlying the figures and results presented in the manuscript been provided?**

Reviewer #1: Yes

Reviewer #2: No: All data underlying the figures are available in the supplement. Some restrictions will apply regarding data access. Data used for the submission may be made available on request to the Twins Early Development Study (TEDS), through their data access mechanism (see www.teds.ac.uk/researchers/teds-data-access-policy).

Reviewer #3: Yes

PLOS authors have the option to publish the peer review history of their article (what does this mean?). If published, this will include your full peer review and any attached files.

Reviewer #1: No

Reviewer #2: No

Reviewer #3: No

---

## [Decision Letter · Decision Letter 1]

15 Sep 2020

Dear Dr Allegrini,

We are pleased to inform you that your manuscript entitled "Multivariable G-E interplay in the prediction of educational achievement" has been editorially accepted for publication in PLOS Genetics, subject to the proposed waiver of data access restrictions we discussed. Congratulations!

Yours sincerely,

Chris Cotsapas, PhD

Associate Editor

PLOS Genetics

Scott Williams

Section Editor: Natural Variation

PLOS Genetics

Comments from the reviewers (if applicable):

Reviewer's Responses to Questions

**Comments to the Authors:**

Reviewer #1: The authors addressed all points that I am concerned.

Reviewer #2: I agree with the authors there are good arguments for such a composite measures, as the SES and chaos mesures applied in the manuscript. They also do appear to predict well.

I do not agree that life events often represent stochastic events unrelated to each other. They tend to correlate and have similar effect on e.g. psychopathology, I guess that could be the case for effect on EA as well, which is in line with the authors findings. Further, some of these life event variables are probably rather rare, and therefore not powerful in separate analyses.

In the G, E model comparisons it makes no difference, but in figure 1C and figure 2, finding significant effect of the composite measures and not of the individual life events is not surprising?

For choices of PRS’s applying the most powerful does make sense, implying this may well change in the coming years. How was power assessed?

I do acknowledge there is no chance adding all the variables that capture EA relevant influences, therefore of course it should be as clear as possible how you choice your variables, and what the limitation is according to that choice.

**Have all data underlying the figures and results presented in the manuscript been provided?**

Reviewer #1: Yes

Reviewer #2: **No: **there are restriction on access, due to confidentiality for participants

PLOS authors have the option to publish the peer review history of their article (what does this mean?). If published, this will include your full peer review and any attached files.

Reviewer #1: No

Reviewer #2: No

**Data Deposition**

http://datadryad.org/submit?journalID=pgenetics&manu=PGENETICS-D-20-00796R1

**Press Queries**

---

## [Editor Report · Acceptance letter]

28 Oct 2020

PGENETICS-D-20-00796R1 

Multivariable G-E interplay in the prediction of educational achievement 

Dear Dr Allegrini, 

We are pleased to inform you that your manuscript entitled "Multivariable G-E interplay in the prediction of educational achievement" has been formally accepted for publication in PLOS Genetics! Your manuscript is now with our production department and you will be notified of the publication date in due course.

With kind regards,

Matt Lyles

PLOS Genetics

On behalf of:
